# Vaginal Laser Therapy for Female Stress Urinary Incontinence: New Solutions for a Well-Known Issue—A Concise Review

**DOI:** 10.3390/medicina58040512

**Published:** 2022-04-04

**Authors:** Alessandro Ferdinando Ruffolo, Andrea Braga, Marco Torella, Matteo Frigerio, Chiara Cimmino, Andrea De Rosa, Paola Sorice, Fabiana Castronovo, Stefano Salvatore, Maurizio Serati

**Affiliations:** 1Obstetrics and Gynecology Unit, IRCCS San Raffaele Hospital, Vita-Salute University, 20132 Milan, Italy; alesruffolo@gmail.com (A.F.R.); salvatore.stefano@hsr.it (S.S.); 2Department of Obstetrics and Gynecology, EOC-Beata Vergine Hospital, 6850 Mendrisio, Switzerland; andrea.braga@eoc.ch (A.B.); castronovo.fabiana@eoc.ch (F.C.); 3Department of Obstetrics and Gynecology, Second Faculty, 80100 Naples, Italy; marcotorella@iol.it; 4ASST Monza, Ospedale San Gerardo, 20900 Monza, Italy; frigerio86@gmail.com; 5Department of Obstetrics and Gynecology, Del Ponte Hospital, University of Insubria, 21100 Varese, Italy; chia.cimmino@gmail.com (C.C.); andreaderosa.unisi@gmail.com (A.D.R.); paodotto@gmail.com (P.S.)

**Keywords:** stress urinary incontinence, urinary incontinence, SUI, vaginal laser, CO_2_ laser, CO_2_ vaginal laser erbium YAG laser, Er:YAG laser, mini-invasive treatment

## Abstract

*Background and Objectives*: Insufficient connective urethra and bladder support related to childbirth and menopausal estrogen decrease leads to stress urinary incontinence (SUI). The aim of this review is to narratively report the efficacy and safety of new mini-invasive solutions for SUI treatment as laser energy devices, in particular, the microablative fractional carbon dioxide laser and the non-ablative Erbium-YAG laser. *Materials and Methods:* For this narrative review, a search of literature from PubMed and EMBASE was performed to evaluate the relevant studies and was limited to English language articles, published from January 2015 to February 2022. *Results*: A significant subjective improvement, assessed by the International Consultation on Incontinence Questionnaire-Short Form (ICIQ-UI-SF) was reported at the 6-month follow up, with a cure rate ranged from 21% to 38%. A reduction of effect was evidenced between 6 and 24–36 months. Additionally, the 1-h pad weight test evidence a significant objective improvement at the 2–6-month follow up. *Conclusions*: SUI after vaginal laser therapy resulted statistically improved in almost all studies at short-term follow up, resulting a safe and feasible option in mild SUI. However, cure rates were low, longer-term data actually lacks and the high heterogeneity of methods limits the general recommendations. Larger RCTs evaluating long-term effects are required.

## 1. Introduction

Stress urinary incontinence (SUI) occurs when the increase in intrabdominal pressure exceeds the urethral closure pressure [1], leading to an involuntary leakage of urine on effort, coughing or sneezing [2,3]. 

Insufficient connective support of the urethra and bladder and the impairment of the urethral sphincter can be responsible for a reduction in the urethral closure pressure [4]. 

Damage of pelvic floor structures due to vaginal delivery and the alteration of collagen composition of the endopelvic fascia, related to the menopausal decrease in estrogen, can lead to an insufficient urethral support [5].

SUI is the most common type of urinary incontinence (UI) [6], occurring in pure or mixed forms in the 80% of incontinent women [7], with a prevalence in middle-aged patients estimated to be among 4–35% [2] and with a huge negative impact on quality of life [7]. 

Conservative treatments, based on lifestyle modification, pelvic floor muscle training (PFMT) and biofeedback/electrical stimulation, are usually adopted as the first-line options, even if these therapies lack long-term compliance and often correct training [8]. 

Mid-urethral slings (MUS) have become the main option and the gold standard for many women affected by SUI [9], due to their high cure rate between 64.1% and 89.1% [10]. The first retropubic tension free vaginal tape (RT-TVT) was introduced by Petros and Ulmsten [11,12]. Despite the improved effectiveness provided by the RT-TVT, adverse events such as bladder perforation, bowel and vessel injuries and voiding dysfunctions [13,14] led De Leval, in 2001, to ideate the trans-obturator (TVT-O) approach [15], being able to reduce these complications but burdened by post-operative groin pain and voiding dysfunction [16]. Since that time, many other devices have been developed, with a variety of introducer mechanisms or mesh types, with an overall rate of reoperation for sling revision or removal of 1.1% [17].

Several pharmacological treatments have also been investigated without reaching a sufficient effectiveness, and an ideal pharmacological agent to manage SUI is lacking [18].

The objective to reduce the invasiveness of anti-incontinence procedures, principally after the American Food and Drug Administration (FDA) warning announcement regarding the use of transvaginal mesh and slings [19], determined the spread of the mini-invasive typology of surgery as urethral bulking agents (UBA) [20,21] and energy-based therapies.

Urethral bulking involves the injection of agents into the submucosal tissues of the urethral or bladder neck to improve urethral closure pressure, increasing the resistance to urine flow. UBAs are demonstrated to be safer than other surgical procedures for SUIs; when compared with MUS, the latter demonstrate a significantly better objective cure rate [22], but with a similar subjective satisfaction rate. 

In this climate of predominance of mini-invasiveness in order to reduce complications, energy-based therapies have become one of the main fields of research in UI. 

Radiofrequency ablation, classified into laparoscopic, transurethral or transvaginal, due to its characteristic of the denaturation of collagen and shrinkage of endopelvic fascia, is nowadays used to treat SUI, with a variable success rate [22,23,24,25,26,27,28,29]. 

Among vaginal lasers, three different laser modalities have been described for treating SUI: the microablative fractional carbon dioxide (CO_2_) laser therapy (10,600 nm) [30]; the dual-phase erbium-doped yttrium aluminium garnet (Er:YAG) laser therapy (2940 nm) combining fractional cold ablation and thermal ablation [31]; and the non-ablative Er:YAG laser therapy (2940 nm) with SMOOTH mode technology [32,33]. In all three cases, laser therapy induces the denaturation of collagen and subsequent neocollagenesis, which is followed by the thickening and strengthening of the anterior vaginal wall, and could determinate a greater support of the bladder and urethra and is, consequently, an improvement of the continence [30,34]. 

Consequently, the purpose of this review is to narratively report, primarily, the subjective and objective efficacy, and secondarily, the safety of Er:YAG and CO_2_ laser energy devices for women affected by SUI.

## 2. Laser Functioning and Rationale in Stress Urinary Incontinence

Vaginal lasers have been originally developed and are demonstrated to be safe and effective in the treatment of vulvovaginal atrophy (VVA) symptoms related to the genitourinary syndrome of menopause (GSM) [35,36]. GSM includes a variety of symptoms due to the decrease in estrogen, resulting in changes of the lower urogenital tract [37]. Estrogens decrease is associated with urogenital symptoms such as SUI, urgency, dysuria and recurrent lower urinary tract infection (UTI), and this is demonstrated by “physiological” data, such as the existence of hormonal receptors (for estrogens and progestins) in the epithelial tissues of the bladder, urethra and trigone and also in the vagina, uterosacral ligaments, levator ani and pubo-rectal muscles [38]. Indeed, estrogen increases the trophism of the epithelial cells of the vagina, urethra and bladder, the peri-urethral vascularization (an important factor involved in the regulation of closing pressure) and the urethral maximum closing pressure [39,40,41]. 

Moreover, multiparous women commonly complain of SUI. In fact, childbirth trauma leads to fascial injury that becomes more elastic and vulnerable, principally in antenatal symptomatic women [42].

Laser therapies determinate the denaturation of collagen fibers and subsequently achieve the remodeling of subepithelial connective tissue and fascia [43], leading to an improvement of strengthened sub-urethral and pubo-cervical fibromuscolaris and fascial support, consequently achieving an improvement of SUI symptoms.

The two main typologies of laser developed and adopted for SUI treatment are the microablative fractional carbon dioxide (CO_2_) laser (10,600 nm) and the non-ablative Er:YAG laser (2940 nm) with SMOOTH mode technology. 

The CO_2_ laser penetrates to a 600 μm of depth and with its ablative effect leads to protein coagulation and tissue necrosis, promoting fibroblast proliferation and consequently neocollagenesis and neoangiogenesis [30]. The Er:YAG laser leaves the vaginal epithelium intact, reaching 500 μm of depth [32,33], with a non-ablative effect due to its major affinity for water absorption that determinates a deeper and more controlled heating of the target, promoting collagen remodeling [43]. 

A temperature of 60–70 °C in the supporting tissue of the urethra [32] induces the denaturation of dermal collagen, collagen remodeling and new collagen formation [44]. 

Moreover, histological studies on vaginal wall biopsies after treatment reported for both types of laser the increase in epithelium thickness, increased number, volume density and diameter of capillaries, enlargement of stromal papillae and collagen remodeling with the reorganization of the fibrillar structures of the extracellular matrix [34,45]. 

## 3. Literature Research

For this narrative review on the efficacy and safety of vaginal laser treatments (including the CO_2_ laser and erbium YAG laser) for SUI, a search of the literature from PubMed and EMBASE was performed in order to evaluate relevant studies. The following key words were used for the research: “stress urinary incontinence”, “urinary incontinence”, “incontinence”, “lasers”, “laser”, “vaginal laser”, “CO_2_ laser”, “CO_2_ vaginal laser”, “erbium”, “erbium YAG”, “er:YAG” and “erbium YAG laser”. 

Two reviewers independently screened the titles and abstracts of the articles. When the opinion between reviewers was divergent, a third author resolved the discrepancy. The evaluated information was the type of article, sample size, type of laser adopted, type of urinary incontinence, menopausal status, number of laser sessions, time to follow up, subjective and objective outcomes and adverse events.

The types of study included were prospective randomized or non-randomized clinical trials, retrospective studies and pilot studies; contrarily, the systematic review, metanalysis, case series, case report, comments, letters, animal studies and pre-clinical and basic research experiments were excluded. The search was limited to English language articles published from January 2015 to February 2022.

The patients included were women affected by SUI based on the diagnostic criteria of the International Continence Society.

The evaluated interventions were the vaginal Erbium YAG laser and CO_2_ laser therapy.

The primary outcomes evaluated were the subjective improvement assessed by the International Consultation on Incontinence Questionnaire-Short Form (ICIQ-UI-SF) and the objective improvement assessed by the changes in the 1-h pad test score. The secondary outcome was to evaluate the nature of the reported adverse events.

## 4. Characteristics of Included Studies

Ninety-three studies from PubMed and 406 from EMBASE were selected for this review. The screening for duplicated papers led to 447 documents. According to the inclusion and exclusion criteria, another 398 studies were excluded. Other 30 articles were reviewed and excluded because they did not satisfy the eligibility criteria. The studies concerning the efficacy and safety of laser treatment for SUI included in this narrative review were 19. Most of the them are prospective non-randomized cohort studies [30,42,46,47,48,49,50,51,52,53,54,55] that were rarely controlled [56,57], with few randomized controlled trials (RCTs) [58,59]; others have a retrospective design [60,61,62]. 

The sample size among the studies ranged from 19 to 84 patients, and only a few studies reported more than 100 women enrolled [30,54,56].

Most of the literature is based on the Er:YAG results, while a small number of articles assessed the efficacy of the microablative fractional CO_2_ vaginal laser for SUI [30,49,59,62]. 

The number of sessions ranged from 1 to 5; however, a cycle of laser therapy often consists in 2–3 laser treatments at 4–6 week intervals. 

More frequently, the studies evaluated only women complaining of isolated SUI with a minor part of them reporting results for the treatment of the stress component of patients affected by mixed urinary incontinence (MUI) [46,47,48,52,54,59,61,62]. The diagnosis of the type of incontinence was clinical or urodynamics-based [47,48,52,58,63].

In women affected by pelvic organ prolapse (POP), a condition of SUI often coexists and, as prolapse advances, women may experience improvement in SUI, but increased difficulty voiding [63]. Almost all studies, in fact, excluded women affected by pelvic organ prolapse higher than the second stage.

Additionally, incontinence onset or severity can be influenced by the GSM [38]: unfortunately, few studies assessed a homogeneous population of only premenopausal [58] or only postmenopausal women [56,57,62].

The results are frequently evaluated in a range of 1- to 6-month follow ups; it was rare for the last follow up to reach 24 [48,56] or 36 months [30]. 

## 5. Subjective Outcomes

The International Consultation on Incontinence Questionnaire-Short Form (ICIQ-UI-SF) is the most used validated questionnaire for subjectively assessing the severity of urinary symptoms, with extremely rare exceptions [52]. This is a six-item questionnaire on a 0–21 scale; the higher the score indicated, the more there are symptoms of urinary distress [64]. 

A significant reduction in median ICIQ-SF scores from baseline to post-intervention was reported for all grades of SUI (from mild to severe), reflecting the efficacy of laser treatment at 6 months of follow up [32,46,47,48,53,55,56,57,60]. An improvement at 6 months follow up was also described in women affected by MUI [46,47,48,54,61], even if it was lower in comparison with women complaining of pure SUI [46,54]. 

The subjective cure rate of women affected by SUI is rarely described. At a ranged follow up of 1 to 6 months, the reported cure rate resulted from 21% to 38% [32,47,50,55,58,60], and a longer subjective cure rate of 62% at the 12-month follow up was also documented [54]. Moreover, a general improvement rate was reported between 75% and 78% at the 6-month follow up [49,50,55,60]. However, the persistence of SUI improvement at the long-term follow up is under-reported: The erb:YAG laser determined an improvement of SUI symptoms at 24 months of follow up, with respect to the baseline, with a reduction of effect between 6 and 24 months [59], while for the CO_2_ vaginal laser therapy there was described a maintenance of the effect at 36 months of follow up, even if reduced between 24 and 36 months [30]. 

Although vaginal lasers therapies were demonstrated to improve SUI from mild to severe conditions, it is reported that in severe forms, reaching success became progressively more difficult [48,58].

Additionally, when the number of sessions was evaluated, it was reported that three laser sessions at 4–6 week intervals achieved greater results in comparison with 1 or 2 sessions; however, no advantages were obtained with more than three laser sessions [30,56,57,60], and some authors recommended annual maintenance treatment [56,57,60].

When the characteristics of the study populations were evaluated, it was resulted that younger women, premenopausal status and lower body mass index (BMI) promoted better results [46,55].

## 6. Objective Outcomes

The 1-h pad weight test was the principal method adopted to objectively assess SUI [65], even if scarcely used among studies [52,53,59,60,62], with, in most of the cases, the evidence of a significant improvement of SUI [48,52,53,61] at a follow up between 2 and 6 months. Usually, a pad test weight less than 2 g is considered as negative [66] with the only reported objective cure rate of 39% and objective improvement rate of 78% at the 6-month follow up [52].

The histological finding of punch biopsies obtained at the urethro-vesical junction in the anterior compartment at the end of the erb:YAG treatment protocol revealed a thicker epithelium with a higher population of intermediate and shedding superficial cells and underlying connective tissue with papillae indenting the epithelium–connective tissue junction at 6 weeks from the last laser sessions; these histological changes were resulted to be related to the subjective and objective improvement of SUI [30].

## 7. Adverse Events

On 30 July 2018, the United States Food and Drug Administration (FDA) issued a warning against the use of energy-based devices (EBDs) to perform vaginal rejuvenation or vaginal cosmetic procedures [19]. However, no significant adverse event was reported in any of the evaluated studies. The events described were principally of minor entity as vaginal discharges, vaginal discomfort/pain during the procedure, vaginal burning and vaginal bleeding/spotting. However, all these adverse events healed without medical interventions and in few days after the procedure.

## 8. Limitations of the Included Studies

The majority of the study designs are prospective, non-randomized and non-controlled low-quality investigations, and few RCTs have been [58,59] produced. There are no papers to date comparing CO_2_ or Erbium YAG lasers for treating SUI with the standard of care or traditional surgeries. 

Looking at the characteristics of the study populations, low sample size is often reported [41,47,48,49,50,52,53,57,58,59,60,61,62], and rarely a pre-study power analysis has been performed [58]. Moreover, the populations enrolled frequently were resulted heterogeneous in relation to the menopausal state and exclusion/inclusion criteria. Indeed, only a few studies excluded women undertaking estrogen therapies [56,61,62], as it is well known that estrogen can impact on urinary symptoms principally in women affected by GSM [67]. Furthermore, BMI is a well-known risk factor for the development of SUI and for anti-incontinence procedure failure [68], and elevated BMI has been sometimes used as exclusion criteria [32,48,55]; this can be considered a selection bias of the enrolled populations. 

Diagnosis of the typology of urinary incontinence and severity assessments were also deficient. In fact, the urodynamic test is an objective method to diagnose a urinary incontinence pattern and can be useful in the preoperatory evaluation, as well as in women complaining of pure SUI, due to the possibility of reducing the number of “useless” anti-incontinence interventions [69]. However, when the study procedures are evaluated, only a few of them objectively assessed the lower urinary tract function with a urodynamic analysis before the treatment [47,48,52,58,61]; furthermore, the severity of SUI, when reported, was differently defined among the studies. Additionally, some studies included patients affected by MUI [46,47,48,52,54,59,61,62], but sometimes did not distinguish the data of women with SUI from the overlap of urgency symptoms [52,59,61,62], leading to hardly interpreted conclusions.

Laser setting protocols presented large heterogeneity, principally among erb:YAG studies, and a different number of treatment sessions are reported. The time of follow up is heterogeneous as well, and often short-term.

Although almost all studies adopted the ICIQ-UI-SF to assess subjectively urinary symptoms, a cure/improvement rate has been rarely reported [32,47,50,54,55,58,60] and the objective evaluation of urinary distress has been occasionally assessed [48,49,52,53,61].

## 9. Conclusions

One of the actual goals in urogynecology is the research of minimally invasive techniques to treat SUI.

SUI after vaginal laser therapy was statistically improved in almost all studies at the short-term follow up of 6–12 months but improvement diminished at 24–36 months. Furthermore, the observed cure rates were quite low and the longer-term data is lacking. Although the reported safety and minimally invasive approach make laser therapy a safe and feasible option in mild SUI, in order to avoid more invasive procedures, the high heterogeneity of the protocols and procedures of inclusion/exclusion criteria and of adopted outcomes limit general recommendations.

When a therapy is proposed to a patient, it is crucial to remember the level of evidence reported. This is why larger RCTs evaluating the long-term effects of laser treatment are required to further investigate its potential harm and efficacy in the treatment of SUI.

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
