# Peer review of "Vaginal Laser Therapy for Female Stress Urinary Incontinence: New Solutions for a Well-Known Issue—A Concise Review"

_medicina, 2022, doi:10.3390/medicina58040512_

Round 1
Reviewer 1 Report
This study is a critical review of the literature related to LASER treatment of female stress urinary incontinence. It should be clearly presented in the Title.
PRISMA statement should be followed for the critical reviews.
PICO principals (Participants, Interventions, Comparators, Outcomes) should be used to for Aims and Goals of the study.
All above mentioned will allow to synthetize a solid conclusion
Author Response
This study is a critical review of the literature related to LASER treatment of female stress urinary incontinence. It should be clearly presented in the Title.
PRISMA statement should be followed for the critical reviews.
PICO principals (Participants, Interventions, Comparators, Outcomes) should be used to for Aims and Goals of the study.
All above mentioned will allow to synthetize a solid conclusion.
Thank you for your suggestions. The type of article that was assigned us was a narrative review. That is the reason why a systematic methodology lacks, even if we have tried to write this review according with some of “systematic” rules. We’ll try to improve the manuscript answering to the PRISMA statements and to the PICO principals.
Reviewer 2 Report
The research of the present article is expensive and it is proven by its numerous bibliography.
The chosen subjects is rich and may be found very easy in the medical literature, therefor I consider that a systematic review or a meta-analysis would be more relevant than a concise review. The conclusions are more accurate from a systematic review and the quality of the final manuscript is superior.
I recommend a reorganisation of the methodology in order to obtain more precise results and conclusions. The discussion part can be very well sustained with the present manuscript.
Author Response
The research of the present article is expensive and it is proven by its numerous bibliography.
The chosen subjects is rich and may be found very easy in the medical literature, therefor I consider that a systematic review or a meta-analysis would be more relevant than a concise review. The conclusions are more accurate from a systematic review and the quality of the final manuscript is superior.
I recommend a reorganisation of the methodology in order to obtain more precise results and conclusions. The discussion part can be very well sustained with the present manuscript.
Thank you for your comment. We have tried to reorganize the methodology using PRISMA and PICO rules, even if the methodology that was assigned to our article was to narratively review our topic.
Reviewer 3 Report
I congratulate you for coming up with this article. Level of Originality, Writing Style and Clarity is good in this article. I thank the authors for the great manuscript and the subject they decided to shed light on.
- It is scientifically sound and contains sufficient interest and originality to merit publication.
Author Response
I congratulate you for coming up with this article. Level of Originality, Writing Style and Clarity is good in this article. I thank the authors for the great manuscript and the subject they decided to shed light on.
It is scientifically sound and contains sufficient interest and originality to merit publication.
Thank you for the revision of the manuscript and for your consideration.
Round 2
Reviewer 1 Report
the manuscript could be accepted in present form